# Nanocomposites for X-Ray Photodynamic Therapy

**DOI:** 10.3390/ijms21114004

**Published:** 2020-06-03

**Authors:** Zaira Gadzhimagomedova, Peter Zolotukhin, Oleg Kit, Daria Kirsanova, Alexander Soldatov

**Affiliations:** 1The Smart Materials Research Institute, Southern Federal University, 344090 Rostov-on-Don, Russia; dkirs27@gmail.com (D.K.); soldatov@sfedu.ru (A.S.); 2Academy of Biology and Biotechnology, Southern Federal University, 344090 Rostov-on-Don, Russia; pvzolotuhin@sfedu.ru; 3Department of Oncology, National Medical Research Centre for Oncology, 344037 Rostov-on-Don, Russia; onko-sekretar@mail.ru

**Keywords:** X-ray photodynamic therapy, photodynamic therapy, photosensitizer, scintillating nanoparticle, reactive oxygen speeches, cancer

## Abstract

Photodynamic therapy (PDT) has long been known as an effective method for treating surface cancer tissues. Although this technique is widely used in modern medicine, some novel approaches for deep lying tumors have to be developed. Recently, deeper penetration of X-rays into tissues has been implemented, which is now known as X-ray photodynamic therapy (XPDT). The two methods differ in the photon energy used, thus requiring the use of different types of scintillating nanoparticles. These nanoparticles are known to convert the incident energy into the activation energy of a photosensitizer, which leads to the generation of reactive oxygen species. Since not all photosensitizers are found to be suitable for the currently used scintillating nanoparticles, it is necessary to find the most effective biocompatible combination of these two agents. The most successful combinations of nanoparticles for XPDT are presented. Nanomaterials such as metal–organic frameworks having properties of photosensitizers and scintillation nanoparticles are reported to have been used as XPDT agents. The role of metal–organic frameworks for applying XPDT as well as the mechanism underlying the generation of reactive oxygen species are discussed.

## 1. Introduction

Currently, cancer treatment is an increasingly important issue. A comparatively new method known as X-ray photodynamic therapy is considered to be a promising one. In order to understand how X-ray photodynamic therapy (XPDT) works, first one needs to analyze some basic principles of conventional photodynamic therapy. In turn, photodynamic therapy has long been studied by scientists around the world. The principle of standard photodynamic therapy (PDT) is widely described in the existing literature [1,2,3,4,5,6,7,8,9,10,11,12,13,14,15,16,17,18,19]. In particular, the basic principle of PDT has been reported [1,2,11,12,16,17]: the Cerenkov Radiation mechanism [4], a detailed consideration of scintillating nanoparticles and their role in the activation processes of photosensitizers [3,6,7,10,14,15,17,19,20] as well as their medical applications can be found in [21,22].

Photodynamic therapy is a clinically approved, minimally invasive therapeutic approach that can exert a selective cytotoxic activity toward malignant cells. PDT applies drug nanoparticles, called photosensitizers or photosensitizing agents and a specific value of light wavelength. Photosensitizers (PSs) are introduced into problem cells and then they are exposed to a definite light wavelength. Upon the completion of the irradiation process PSs generate some forms of oxygen that destroy the vascular system of cancer tissues (Figure 1). Each PS is activated by a defined light of wavelength and the penetration of this light depends on the value of the wavelength. Thus, specific photosensitizers, scintillating nanoparticles (ScNP) and light wavelengths are used for deeper penetration [8,9,12].

Tumors deep inside the body are not usually treated with photodynamic therapy. A relatively new approach, XPDT [10,11,23,24,25,26,27,28,29,30,31], is based on X-rays which can penetrate more deeply into tissue than PDT light wavelength. XPDT is much better than traditional PDT (excited by UV–Vis light) in terms of tissue penetration. 

Most PSs can only be excited by UV–Vis light for treating skin, lungs or esophagus cancer. However, there are photoconverting nanoparticles based on the Förster resonance energy transfer (FRET) [32]. These nanoparticles transform X-rays into UV–Vis light. Then, photosensitizers can produce energy for generating reactive oxygen species (ROS). Such nanoparticles can be used in various combinations with photosensitizers. Not all the combinations are suitable for treatment directly in the body due to their toxicity. Here, different types of biocompatible cytotoxic drug nanoparticles are introduced (Figure 2).

It is expected that XPDT will overcome the limitations of conventional PDT and provide high-precision treatment of deep tumor cells with minimal damage to healthy tissues. 

## 2. Nanoparticles Used in X-Ray Photodynamic Therapy

### 2.1. Types of Photosensitizers

Cu–Cy. Copper–cysteamine Cu–Cy nanoparticles were used as a photosensitizer in [33,34]. 

Shi Lei, Liu Pei et al. found that without X-ray irradiation, copper–cysteamine nanoparticles were non-toxic for keratinocyte cells. Squamous-cell carcinoma was more sensitive to XPDT than melanoma cells. XPDT successfully reduced the growth of squamous cell carcinoma in vivo, while melanoma was stable. Micro-vessel density in squamous-cell carcinoma tissues was remarkably reduced. No obvious acute toxicity was observed. Cells were irradiated by X-ray for 1.5 Gy.

S. Shrestha et al. [34] demonstrated that Cu–Cy nanoparticles may conjugate to a pH-low insertion peptide to ease the active targeting of these nanoparticles to low pH tumors in the future. 

Their results show a significant decrease in tumor size under the X-ray radiation of pH-low peptide-conjugated, copper–cysteamine nanoparticles in tumor-bearing mice. This work confirms the effectiveness of copper–cysteamine nanoparticles as a photosensitizer when activated by radiation and approves that these Cu–Cy nanoparticles are good candidates for PDT in deep tumors combined with X-rays and conjugated to a tumor-targeting molecule. After injecting the nanoparticles, the bodies of the mice were irradiated for 30 min, with the irradiation dose being 5 Gy.

TiO_2_:Ce. Single TiO_2_ is not effective enough in the generation of ROS. For example, in [35] Chun-Chen Yang and his colleagues doped the anatase lattice of TiO_2_ with Ce, which was more effective than a single TiO_2_ for better photosensitivity and ROS generation results. 

Cerium was used to doped the TiO_2_ anatase structures to increase photochemical reactions owing to its strong catalytic potential and light response extension [31]. Furthermore, it could suppress the recombination of electron–hole pairs and increase their lifetime. Additionally, Ce has a greater X-ray photon interaction cross-sectional area than that of biological tissues, which leads to the intense X-ray interaction with control materials and ROS generation [35]. The X-ray radiation dose was approximately 0.133 Gy for 100 s.

TiO_2_:C. Chun-Chen Young and his co-authors [36] investigated nanoparticles with a narrow bandgap and ROS generation under the influence of soft X-ray radiation to destroy tumor cells with less damage to healthy tissues. In this experiment, anatase lattice was doped with C, and then it was activated by X-rays to produce more reactive oxygen species and destroy tumor cells. After X-ray radiation and the introduction of TiO_2_:C to tumor cells, the size of the tumor gradually reduced. The X-ray radiation dose was approximately 0.133 Gy for 100 s.

(n-Bu4N)_2_(Mo_6_I_8_(OCOCF_3_)_6_). Kaplan Kirakci et al. [37] studied the nanoparticles made of the octahedral molybdenum cluster compound or (*n*-Bu_4_N)_2_(Mo_6_I_8_(OCOCF_3_)_6_). These nanoparticles generate singlet oxygen O_2_(^1^Δ_g_) from blood under X-ray irradiation. The (*n*-Bu_4_N)_2_(Mo_6_I_8_(OCOCF_3_)_6_) nanoparticles were prepared by nanoprecipitation. 

(*n*-Bu_4_N)_2_(Mo_6_I_8_(OCOCF_3_)_6_) nanoparticles efficiently absorbed X-rays because they consist of heavy elements, which in turn leads to the generation of the excited triplet states of photosensitizer interacting with molecular oxygen to produce singlet oxygen O_2_(^1^Δ_g_). 

Some specific concentrations of the obtained nanoparticles are non-toxic and quite stable in water. The simple structure of nanoparticles leads to their significantly higher activity at lower concentrations. This fact is mainly based on the energy transfer between scintillating nanoparticles and photosensitizers. In addition, the small size of the nanoparticles provides a large energy transfer. The smaller the particles are, the closer they can approach the target. This means that the value of the transferred energy from the nanoparticle to the molecular oxygen is higher. As a result, it leads to the formation of a higher number of ROS. Thus, the enhancement effectiveness of (*n*-Bu_4_N)_2_(Mo_6_I_8_(OCOCF_3_)_6_) nanoparticles could be achieved by their size. The X-ray dose also was 1 Gy and 2 Gy for 100 s.

Pollen-structured gold cluster. It is worth considering another interesting substance that was well studied and synthesized in [38]. This is the core of the pollen-structured gold cluster (PSGC).

PSGCs were made of mesoporous silica and then their core was passivated by a rich layer of gold nanoparticles (Figure 3). PSGCs were synthesized with a deposition–precipitation process. The architecture of PSGCs provides a large surface area, which in turn leads to the generation of ROS under X-ray irradiation. 

The obtained nanoparticles were exposed to a variety of irradiation doses: 0.5, 1, 2, and 5 Gy. A single dose of X-ray irradiation was provided with the voltage of the tube equal to 160 keV.

PSGCs have a significantly higher ability to generate ROS and they offer a better treatment of breast cancer cells [38], demonstrating the potential clinical application in the treatment of deep tumors.

### 2.2. Types of Scintillating Nanoparticles

Y_2_O_3_:Eu and Y_3_A_l5_O_12_:Eu. Yang C. et al. in [39] investigated the luminescent properties of Y_2_O_3_:Eu and Y_3_A_l5_O_12_:Eu particles as scintillating nanoparticles irradiated with X-rays in PDT. The nanoparticles can be used for cancer treatment in combination with some kind of photosensitizers that selectively accumulate in tumor cells. Two sets of finely dispersed Y_2_O_3_:Eu and Y_3_A_l5_O_12_:Eu phosphors were synthesized by the Pechini method and self-propagating high-temperature synthesis. The data in [39] show that the use of this X-ray activated phosphor in combination with the photosensitizers results in the destruction of tumor cells under irradiation thus confirming a high efficiency of the suggested approach.

Y_2_O_3_. Jonathan P. Scaffidi et al. in [40] investigated the combination of yttrium oxide (Y_2_O_3_) scintillating nanoparticles, a part of the HIV-1 TAT (Human immunodeficiency viruses – 1 trans-activator of transcription) peptide, and psoralen was investigated by X-ray radiation. The X-ray radiation is absorbed by the Y_2_O_3_ nanoparticles, which then emit UV light. The absorption of UV photons by the nanoparticles connected with psoralen has the potential to cross-link adenine and thymine residues in DNA. As is well known, UV-induced cross-linking by free psoralen upon the activation of UV light causes apoptosis and an immunogenic response. Investigations demonstrate that the X-ray irradiation of these psoralen-functionalized Y_2_O_3_ nanoscintillators yields a decrease in the cell number concentration when compared to control cell cultures containing psoralen-free Y_2_O_3_ nanoscintillators. Cell culture were exposed to 2 Gy X-ray radiation with the voltage of tube −160 or 320 kVp.

NaGdF_4_: Gd^3+^, Eu^3+^. Some lanthanide-doped nanoparticles can absorb X-ray radiation. There was a series of Gd^3+^ and Eu^3+^ compositions in lanthanide fluorides for optimizing the emission from Eu^3+^ under X-ray excitation [41]. In addition, the optimum concentration of Eu^3+^ that produced the most intense emission in NaGdF_4_ was 15% molar concentration. Moreover, an attempt to include a sensitizer (Ce^3+^) in NaGdF_4_:Eu^3+^ resulted in a reduction in the emission following X-ray excitation. Moreover, a surface coating of NaGdF_4_:Eu^3+^ nanoparticles with a gold shell decreased the X-ray luminescence by a factor of two in comparison to uncoated phosphors. NaGdF_4_ nanoparticles were doped with Eu^3+^ and Ce^3+^ and were obtained by a citrate method.

LiGa_5_O_8_:Cr. Hongmin Chen in [42] used nanoscintillators based on LiGa_5_O_8_:Cr, which emit persistent near-infrared luminescence. This allows to obtain optical images of deep tissues that can be used to control exposure. In particular, LiGa_5_O_8_:Cr nanoparticles and a 2,3-naphthalocyanine photosensitizer were encapsulated in mesoporous silica nanoparticles. Nanoconjugates can be efficiently accumulated in lung tumors, as evidenced by monitoring X-ray luminescence from LiGa_5_O_8_:Cr. It should be noted that LiGa_5_O_8_:Cr nanoparticles were prepared by a polystyrene phere-assisted sol–gel method.

### 2.3. Combination of Photosensitizers and Nanoparticles

AIE-Au. Wenjing Sun et al. in their work [43] investigated such conjugates of photosensitizers as aggregation-induced emission heterogeneous Au clustoluminogens (AIE-Au) which consist of glutathione-protected gold clusters (GCs) to achieve efficient low-dose XPDT. 

When irradiated, AIE-Au strongly absorbed X-rays to generate hydroxyl radicals, which increased the radiotherapy effect by damaging DNA. Furthermore, the aggregates of glutathione-protected GCs increased the X-ray excited luminescence. The AIE-Au transformed X-rays into optical luminescence and excited the rose bengal (RB) photosensitizers, which oxidizes lipid membranes. In other words, AIE-Au clustoluminogens triggered the generation of reactive oxygen species. X-ray doses used in this investigation were 1 Gy. 

ADH-1-HA-MTN. Zhaoming Guo and others in their investigation in [44] used ADH-1-HA-MTN (Alcohol dehydrogenase-1-hyaluronic acid-mesoporous titanium dioxide nanoparticles) nanoparticle conjugates as drug delivery to tumor tissues. These conjugates are based on mesoporous titanium dioxide nanoparticles (MTN). HA and ADH-1 are attached to the surface of the MTN as an active targeting ligand to target drug delivery to tumor cells. ADH-1 acted as an antagonist to block the function of N-cadherin, which may result in the disruption of tumor vasculature. Moreover, MTN generated a great number of ROS when irradiated. X-ray irradiation was carried out at 60–75 kV and 0.15–0.35 mA. The average size of these conjugates was about 110 nm.

LiYF_4_@SiO_2_@ZnO. Qien et al. in [45] reported herein on the integration of a scintillator and a semiconductor as an XPDT agent. These conjugates are core-shell Ce^III^-doped LiYF_4_@SiO_2_@ZnO nanoparticles (LSZNPs).

In this structure, the ultraviolet fluorescence from the Ce^III^-doped LiYF_4_ nanoscintillator under irradiation gives rise to the formation of electron–hole pairs in ZnO nanoparticles leading to the formation of biotoxic hydroxyl radicals. 

This method demonstrates a reduced dependence on the intracellular oxygen levels by integrating the scintillator and semiconductor as a photosensitizer. These nanoparticles have been used as the scintillation nanoparticles in the down-conversion of X-rays to match the energy gap of a semiconductor to produce ROS from water molecules for an effective explosion.

The volume of the tumor (the HeLa cell line was used) with the introduced LSZNPs and exposed to an X-ray radiation of 8 Gy was almost completely inhibited in 15 days. In addition, late-stage apoptosis featuring karyopyknosis, karyorrhexis and the significant formation of apoptotic bodies was observed after the irradiation of these tissues. Thus, the application of these nanoparticles demonstrates antitumor therapeutic effectiveness. X-ray irradiation was used at 3 Gy.

RGD-ZSM-RB. Zn- and Mn-incorporated silica (ZSM) nanoscintillators were conjugated with photosensitizer rose bengal and arginylglycylaspartic acid (RGD) peptide, due to the fact that they are intrinsically dual modal targeted imaging probes as it was shown in [46]. The proposed RGD-ZSM-RB nanosensitizer shows excellent deep-tumor therapy with low dose X-ray irradiation (1 Gy). After the conjugation of RGD peptides with silicate scintillators, the nanosensitizer could show effective accumulation in cancer cells and local inhibition. Moreover, Zn and Mn dopants are significant elements in the human body. XPDT treatment therefore minimizes the potential adverse effects of local radiotherapy due to the application of low radiation doses like 1 Gy, while the typical dose for a solid epithelial tumor varies from 60 to 80 Gy.

LaF_3_. Lanthanum fluoride is a transparent material that could allow radioluminescence at a set of wavelengths through the simple substitution of lanthanum ions with other luminescent lanthanides. For example, in [47] Kudinov K. et al. prepared lanthanum fluoride nanoparticles doped with cerium, terbium, or both, which had good spectral overlap with Rose Bengal photosensitizer molecules. 

LaF_3_:Tb. Another interesting scintillating nanoparticle was investigated in [48]. They were Tb^3+^-doped LaF_3_ scintillating nanoparticles (LaF_3_:Tb) combined with meso-tetra(4-carboxyphenyl)porphyrin photosensitizer, followed by activation with soft X-rays. Scintillating LaF_3_:Tb nanoparticles were obtained, sized approximately 25 nm. They had good dispersibility in aqueous solutions and demonstrated great biocompatibility. 

Mesoporous LaF_3_:Tb ScNPs were synthesized in the light hydrothermal process by Yong’an Tang, Jun Hu et al. in [32]. LaF_3_:Tb ScNPs act as a photosensitizer and these nanoparticles convert X-ray energy due to their ability to absorb ionizing radiation and great luminescence efficiency. LaF_3_:Tb was obtained with optimized scintillation luminescence for rose bengal. It is necessary to design the effectiveness of the FRET system due to the perfect spectrum matching of LaF_3_:Tb and rose bengal. FRET effectiveness between LaF_3_:Tb and RB was 85%. Under X-rays, an increase in ^1^O_2_ formation was detected due to the induced LaF_3_:Tb-RB nanocomposites during the FRET process. This FRET LaF_3_:Tb-RB system demonstrated great potential for the future application of X-rays in deep tumors.

Studies of Ahmed H. Elmenoufy et al. in [25] also confirm the effectiveness of these conjugates. They investigated multifunctional LaF_3_:Tb coated with homogeneous silicon dioxide layers and covalently bound to rose bengal.

CeF_3_:Tb^3+^. In the studies of K. Popovich et al. [49] CeF_3_:Tb^3+^-based nanoparticles modified with SiO_2_ and protoporphyrin IX (PpIX) were investigated. Nanopowder CeF_3_:Tb^3+^ was prepared via the sol–gel method, with further surface coating by SiO_2_ layer and the conjugation with photosensitizer PpIX. Radioluminescence spectra showed an energy transfer from Ce^3+^ to Tb^3+^ ions and from Tb^3+^ to the molecules of PpIX photosensitizer. The samples were exposed with X-rays using a tube with a Cu anode (voltage 40 kV, current 30 mA, average wavelength K_α1,2_ = 0.15418 nm). 

Emission bands of Tb^3+^ overlap well with the absorption lines of protoporphyrin PpIX. Therefore, the efficient energy transfer from donor to acceptor and the production of singlet oxygen were expected. This combination of nanoparticles generated ROS. Nanocomposites under investigation may be quite good candidates for the application in XPDT.

CeF_3_:Tb^3+^, Gd^3+^. Another promising nanoparticle is CeF_3_ doped with Gd^3+^ and Tb^3+^ (CGT) investigated in [50]. CGT nanoparticles were synthesized with a hydrothermal process. Co-doped CeF_3_:Gd^3+^, Tb^3+^ scintillating nanoparticles were then coated with mesoporous silica. Such doping with Gd and Tb boosted the scintillation properties of the CeF_3_ nanoparticles.

The enhancement of the X-ray-excited optical luminescence was due to the fact that the energy levels of Gd^3+^ lie between the activator (Ce^3+^) and the luminescent center (Tb^3+^), giving rise to efficient energy transfer. The irradiation of nanoparticles was carried out with single X-ray dose 3 Gy.

GdVO_4_:Eu^3+^. Kateryna Hubenko et al. in [51] used GdVO_4_:Eu^3+^ nanoparticles. They demonstrated that photosensitizer-generated free radicals and reactive oxygen species in water solutions containing gadolinium orthovanadate and their complexes with methylene blue. 

Due to the effective excitation energy transmission in the conjugates, they could act as prospective X-ray photodynamic agents. However, under X-ray irradiation, the strong OH radicals scavenging by the nanoparticles were observed. Furthermore, the voltage on the used tube was 30 kV (20 mA). 

ZnGa_2_O_4_:Cr (ZGO:Cr/W). Song L. and others reported in [52] on a low-dose X-ray-activated persistent luminescence nanoparticle (PLNP)-mediated PDT nanoplatform for treating deep cancer tumors. The high persistent luminescence and long persistent luminescence time were the key points for effective cancer treatment. In order to achieve this goal, W^VI^-doped ZnGa_2_O_4_:Cr (ZGO:Cr/W) PLNPs were synthesized. In comparison with the traditional ZnGa_2_O_4_:Cr PLNPs, ZGO:Cr/W PLNPs show a higher persistent luminescence intensity and a longer persistent luminescence time after the same irradiation dose. Then, by coupling with PS Zn(II) phthalocyanine tetrasulfonic acid (ZnPcS_4_), the PDT nanoplatform (ZGO:Cr/W–ZnPcS_4_) was constructed. The constant luminescence could continue exciting the coupled PS after the X-ray irradiation was removed, leading to the reduced X-ray dose (~0.18 Gy for 2 min) to minimize the side effects of XPDT.

ZnS:Cu,Co. Like Lun Ma et al. in [53] synthesized copper- and cobalt-doped ZnS (ZnS:Cu,Co) scintillating nanoparticles which were then used in combination with tetrabromorhodamine-123 (TBrRh123) photosensitizer. It was determined that in the conjugates there was an efficient energy transfer from nanoparticles to photosensitizers. As a result, singlet oxygen was generated for the treatment of cancer cells. 

Furthermore, ZnS:Cu, Co nanoparticles have a long X-ray excited afterglow used as a continuous light source to activate XPDT. ZnS: Cu, Co-TBrRh123 X-ray conjugates are very effective for the destruction of cancer cells. Besides ZnS:Cu, Co nanoparticles irradiated by X-rays (2 Gy) can also be used to visualize cells. All this indicates that ZnS:Cu, Co nanoparticles are promising for biomedical applications.

GdEuC_12_ micelles. It should be mentioned that sometimes lanthanide micelles are used as nanoparticles in XPDT. For instance, Slávka Kaščáková et al. designed a liponanoparticle based on GdEuC_12_ micelles including hypericin as a photosensitizer in their hydrophobic core, which provides singlet oxygen production upon X-ray irradiation [30]. The micelles were composed of amphiphilic lanthanide chelates and incorporated hypericin as photosensitizer in the hydrophobic core. The micelles provided an easy way to integrate highly hydrophobic molecules such as typical photosensitizers. Liponanoparticles may be used as carriers for delivering a high payload of the photosensitizer to the sites. It could not be achieved earlier due to their poor water solubility. It was hypothesized that the X-ray irradiation of these micelles could trigger the cascade of specific reactions that allowed to generate reactive oxygen species. Time-resolved laser spectroscopy and singlet oxygen probes were used for this investigation.

Tb_2_O_3._ Tb_2_O_3_ has recently become the center of attention. Anne-Laure Bulin et al. in [54] used Tb_2_O_3_ coated with a polysiloxane layer. This nanosystem was combined with porphyrin molecules that were able to generate singlet oxygen, which is a major cytotoxic agent in X-ray photodynamic therapy. This combination was suitable for singlet oxygen formation irradiated with X-ray. 

Table 1 shows the agents collected for dose comparison as well as the cell lines that were used in each study.

### 2.4. Metal–Organic Frameworks as Photosensitizers

Metal–organic frameworks (MOFs) are crystalline porous compounds that consist of organic linkers and metal ions. The demonstrate a great potential in drug delivery applications due to their high surface area and drug-loading capacity, tailorable pore structure as well as their functionality [57,58,59,60,61,62]. Many MOFs have already been used as nanoparticles in photodynamic therapy. However, MOFs are also often used as nanoparticles in X-ray photodynamic therapy. Let us consider several structures of MOFs that turned out to be the most successful in various studies.

Hf-DBB-Ru. Recently nanoscale metal–organic frameworks (nMOFs)-Hf-DBB-Ru have been discovered in [63]. Hf-DBB-Ru is (Hf_6_(µ_3_-O)_4_(µ_3_-OH)_4_(DBB-Ru)_6_)^12+^ where DBB-Ru = bis(2,2′-bipyridine)(5,5′-di(4-benzoato)-2,2′-bipyridine)ruthenium (II) chloride. 

Hf-DBB-Ru was synthesized by solvothermal reaction between HfCl_4_ and H_2_DBBRu with trifluoroacetic acid as the modulator. Hf-DBB-Ru was irradiated with 1, 2, 3, 5 or 10 Gy X-ray.

This MOF was designed from Ru-based photosensitizers. The cationic framework demonstrated strong targeting to mitochondria. Under X-ray irradiation, Hf-DBB-Ru successfully generated hydroxyl radicals from Hf_6_ secondary building units (SBUs) and singlet oxygen from the DBB-Ru photosensitizers. Mitochondria-targeted particles depolarized the mitochondrial membrane to activate the apoptosis of cancer cells, which resulted in the significant regression of colorectal tumor cells. When the volume of the tumor reached 100−150 mm^3^, the tissue was irradiated by light with energy 180 J/cm^2^ (650 nm).

Hf-TCPP. Liu J. et al. in [55] reported on the structure of a nMOF which consists of hafnium (Hf^4+^) and tetrakis (4-carboxyphenyl) porphyrin (TCPP). TCPP is a photosensitizer, and Hf^4+^ could serve as a radio-sensitizer to enhance radiotherapy. 

Hf as the element with high Z could interact with ionizing radiation resulting in photo/auger electrons and then generate reactive free radicals to destroy the vascular system of tumor cells [64]. In addition, a series of experiments were carried out to check the cytotoxicity of HfO_2_ nanoparticles in [65]. It is reported that 3T3 fibroblast cell lines were used. The damage of the cell lines was not observed. Thus, the Hf is a biocompatible chemical element. Moreover, nMOF was coated with polyethylene glycol (PEG) and nMOF-PEG nanoparticles demonstrated great stability in physiological solutions. The investigated cells were irradiated with X-ray dose 6 Gy for 3 min.

Hf-DBP and Hf-TBP. Another important application of MOFs is presented in [66]. Kuangda Lu’s et al. used the Hf nMOFs as X-ray scintillating nanoparticles. The synthesis data are presented in [56]. The energy of the X-ray absorbed by the Hf SBUs was directly transferred to anthracene ligands leading to inelastic photoelectron scattering. 

Two nMOFs: 5.15-di (*p*-benzoato) porphyrin-Hf (DBP-Hf) and 5,10,15,20-tetra (*p*-benzoato) porphyrin-Hf (TBP-Hf) were synthesized from Hf clusters and porphyrin photosensitizing ligands. Hf clusters absorb X-ray photons giving rise to the generation of OH radicals and ^1^O_2_ by photosensitizers. nMOFs were exposed with X-rays from 0 to 1 Gy.

Hf_6_-DBA and Hf_12_-DBA. Kaiyuan Ni et al. in [67] investigated two more nMOF structures based on Hf-Hf_6_-DBA (Hf-Hf_6_-DBA = Hf_6_(μ_3_-O)_4_(μ_3_-OH)_4_(DBA)_6_) and Hf_12_-DBA (Hf_12_-DBA = Hf_12_(μ_3_-O)_8_(μ_3_-OH)_8_(μ_2_-OH)_6_(DBA)_9_), where DBA = 2,5-di(*p*-benzoato) anthracene. These nMOFs were synthesized via solvothermal reactions.

nMOFs were constructed as radio-enhancers by taking advantage of the electron-dense Hf_6_O_4_(OH)_4_ and Hf_12_O_8_(OH)_14_ SBUs as X-ray absorbers to produce ROS. 

Under X-ray irradiation, nMOFs convert energy to anthracene-based bridging ligands, DBA, to emit radioluminescence in the visible spectrum range. For example, in [67] Hf_12_-DBA and Hf_6_-DBA (incubated in CT26 cell line) were irradiated with 2 Gy X-rays dose. It was shown that Hf_12_-DBA gave a much brighter radioluminescence signal with respect to Hf_6_-DBA.

Hf_12_-DBA demonstrated great radio-enhancement over Hf_6_-DBA. It has been suggested that this is due to enhanced X-ray absorption by the electron-dense Hf_12_ clusters and hydroxyl radical diffusion through the porous nanoplates. It should be noted that under γ-rays Hf_12_-DBA also exhibited greater radio-sensitization than HfO_2_ and Hf_6_-DBA.

## 3. Reactive Oxygen Species

Reactive oxygen species (ROS) are chemically reactive chemical species containing oxygen. Examples include peroxides, superoxide, hydroxyl radical, singlet oxygen and alpha-oxygen. There are several sources and corresponding mechanisms of ROS generation. One could classify them into two main groups: endogenous and exogenous sources. ROS are produced during a variety of biochemical reactions outside and within the cell—in the cytosol and all organelles (most notably in the mitochondria, peroxisomes and endoplasmic reticulum). Endogenous sources and mechanisms can be stimulated by external agents—these influences can also be classified as exogenous ROS inducers. The formation of ROS can be stimulated by a variety of agents such as pollutants, heavy metals, tobacco, smoke, drugs, xenobiotics or radiation. One could find an extensive review of the ROS sources and corresponding mechanisms in a recent review by Snezhkina et al. [68].

ROS effects in cancer therapy are ambiguous. Being prominent physiological signaling agents, ROS affect cellular responses and modulate adaptiveness. This is a specifically important feature of cancer since malignant cells have a greater adaptive potential than normal cells [69,70]. More importantly, only extreme levels and acute treatments provoke cell-damaging ROS properties in cancer cells. 

In ROS-dependent anti-cancer therapy, this addresses several issues. 

Firstly, therapeutic ROS may increase the antioxidant capacity of the cells by plain antioxidant signaling [71] and promote cell survival via the NF-kappaB and associated pathways [72,73]. Secondly, the induced signaling consequently leads to metabolic hyper-adaptation [74,75], anoikis refractiveness [76], reinforced epithelial to mesenchymal transition [72], and increased metastasis. Thirdly and most importantly, sub-threshold ROS levels do not affect cancer stem cells due to their inherently increased antioxidant capacity [77] and residing in hypoxic niches [78]. All these challenges have made a great number of ROS-based anti-cancer approaches ineffective [79]. 

ROS-based approaches rely on cancer cell distraction due to the oxidative damage of cellular components induced by ROS donors and redox-cyclers (chemical mode of action) or ROS inducers (signaling mode of action). Although they have been effective against transit-amplifying cancer cells, these do not affect cancer stem cells—which, along the epithelial–mesenchymal transition process, are the primary target in cancer therapy. 

Cancer stem cells possess inherently increased antioxidant capacity and most therapeutic agents (including those inducing ROS) are incapable of prolonged residing in cancer stem cells due to drug efflux pumps highly represented on the membrane of cancer stem cell. Additionally, just as normal stem cells, cancer stem cells localize in hypoxic niches and deficient oxygen makes ROS generation intensity a priori low. Additionally, ROS inducers disregarding their mode of action promote strong antioxidant and anti-xenobiotic signaling further enhancing the cancer stem cell resistance to therapy.

For ROS-based therapy to be effective, treatment has to be acute and powerful [70]. If designed and applied correctly, ROS-based therapy may induce ROS-dependent autophagic death [80], receptor- and mitochondria-mediated apoptosis [70], necrosis [70], DNA damage, parthanatos cell death [81] and secondary signaling ROS generation [70]. The more modalities are combined, the better. 

Photodynamic therapy holds a unique position among ROS-dependent anti-cancer technologies, because it is somewhat selective [82,83], always acute [84], powerful [85] and does not always rely on normoxic conditions [63]. For instance, type I PDT technology is capable of inducing ROS generation even in hypoxia [26,63] thus making cancer stem cells sensitive to treatment along with transit-amplifying cells.

Type I PDT agents are unique in this regard. The dependence of type I PDT on molecular oxygen levels is lower than that of the type II PDT if hydroxyl radical is generated, and consequently, type I PDT allows oxygen-independent ROS generation. Since type I PDT agents do not require oxygen and do not need to enter or remain within a cancer stem cell to induce cell death (and thus, do not induce adaptive signaling changes in the cell), these agents do not have the limitation characteristics of most ROS-based agents for cancer therapy imposed by the above discussed facts.

## 4. Conclusions

To conclude, one can say that the problem of cancer is socially important. The photodynamic therapy method was developed to solve the problem. However, the approach is suitable only for diseases located on the surface of the tissue such as melanoma. Recently, the deeper penetration of X-rays into the tissue has been implemented, which is now known as X-ray photodynamic therapy. In this case, X-ray radiation is used to activate scintillating nanoparticles. Moreover, due to the penetrating power of X-rays, XPDT is applicable for the treatment of deeper cancer tumors. The two methods differ in the photon energy used, thus requiring the application of different types of scintillating nanoparticles. The challenge is to find the most stable, biocompatible, non-toxic and targeted structures for the treatment of deeper cancer tumors.

Therefore, in this article, various types of nanoparticles, photosensitizers and their combinations were considered. For instance, there is an interesting combination of ZnGa_2_O_4_:Cr with ZnPcS_4_ which is activated by low doses of X-ray radiation (~0.18 Gy for 2 min). Moreover, some nanoparticles combine properties of both photosensitizers and scintillating nanoparticles. These nanoparticles are activated by rather low doses of irradiation in the range from 0.133 to 8 Gy (see details in Table 1). For example, TiO_2_:Ce and TiO_2_:C were irradiated with an X-ray dose of 0.133 Gy for 100 s. These doses are much less than in other cases. For instance, (*n*-Bu_4_N)_2_(Mo_6_I_8_(OCOCF_3_)_6_) and ZnS:Cu, Co/TBrRh123 were exposed to 2 Gy. With regard to MOFs, Hf-DBB-Ru nMOFs in the tissue were irradiated by 2 Gy (in energy 180 J/cm^2^ or 650 nm). The heavy metal makes MOFs promising candidates for X-ray photodynamic therapy. However, radiation doses for the activation of MOF structures are higher than the dose for TiO_2_:Ce and TiO_2_:C.

## Figures and Tables

**Figure 1 ijms-21-04004-f001:**
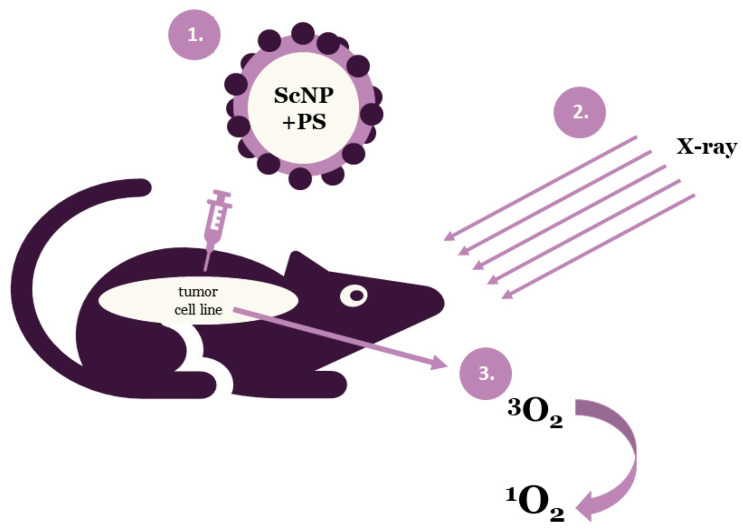
Schematic illustration of the general principle of X-ray photodynamic therapy (ScNP—scintillating nanoparticle, PS—photosensitizer).

**Figure 2 ijms-21-04004-f002:**
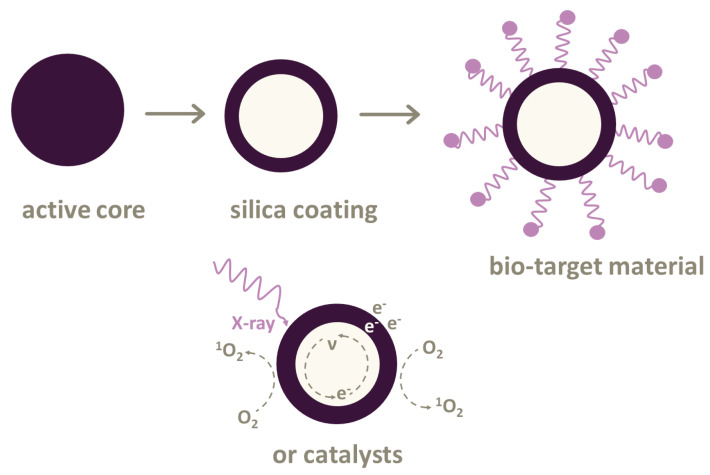
Nanoparticles used in X-ray photodynamic therapy (XPDT) and photodynamic therapy (PDT). The active core of nanoparticles can be coated with silica cover. It is possible to attach linkers or bio-target materials to some nanoparticles or use various types of catalysts.

**Figure 3 ijms-21-04004-f003:**
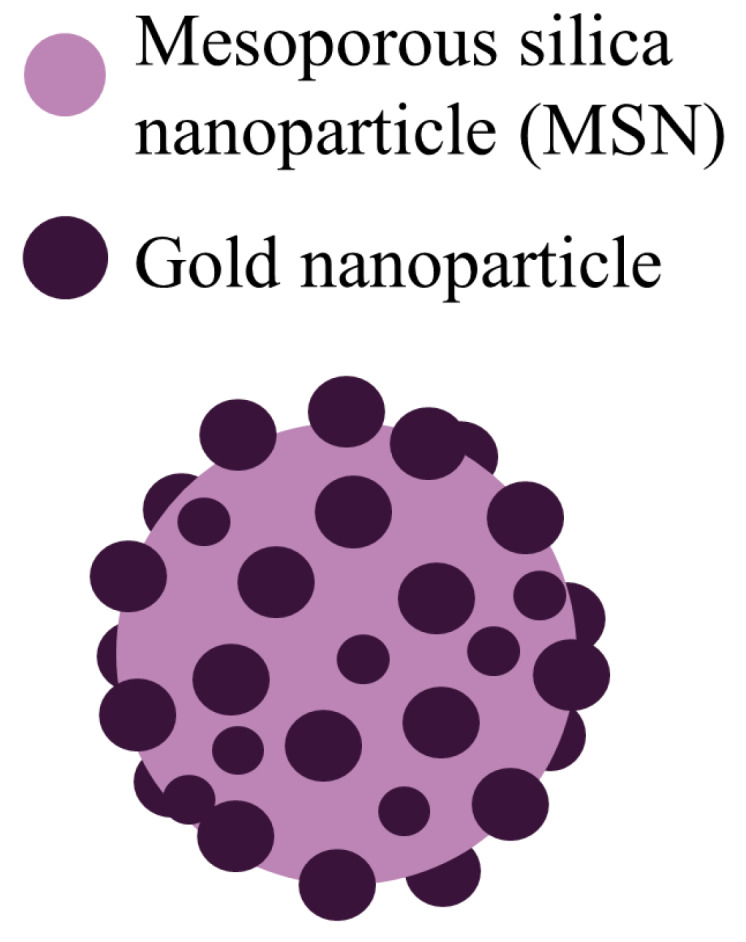
Schematic illustration of the pollen-structured gold cluster. The cluster consists of a mesoporous silica core and a gold active shell.

**Table 1 ijms-21-04004-t001:** Agents used in XPDT: PS without ScNP, PS with ScNP and metal–organic framework (MOF) structures.

Photosensitizer	Scintillating Nanoparticle	Absorbed Dose of Ionizing Radiation (Gy)	Biological Experiments	Reference
Cu–Cy	-	5 Gy for 30 min	Squamous cells carcinoma	[34]
TiO_2_:Ce	-	0.133 Gy for 100 s	A549	[35]
TiO_2_:C	-	0.133 Gy for 100 s	A549	[36]
(*n*-Bu_4_N)_2_(Mo_6_I_8_(OCOCF_3_)_6_)	-	1 and 2 Gy	HeLa and MRC	[37]
RB	CeF_3_:Tb^3+^, Gd^3+^	3 Gy	Mgc803, HEK293T and 4T1	[50]
ZnGa_2_O_4_:Cr	ZnPcS_4_	~0.18 Gy for 2 min	HeLa	[52]
ZnS:Cu, Co	TBrRh123 (tetrabromorhodamine-123)	2 Gy	PC3	[53]
nMOF Hf-TCPP	Hf as radio-sensitizer;	6 Gy for 3 min	4T1, HeLa, and NIH3T3	[55]
nMOF Hf_6_-DBA (Hf_6_(μ_3_-O)_4_(μ_3_-OH)_4_(DBA)_6_) and Hf_12_-DBA (Hf_12_(μ_3_-O)_8_(μ_3_-OH)_8_(μ_2_-OH)_6_(DBA)_9_)	Hf as radio-sensitizer	2 Gy	CT26	[56]

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
