# Peer review of "Nanocomposites for X-Ray Photodynamic Therapy"

_ijms, 2020, doi:10.3390/ijms21114004_

Round 1

Reviewer 1 Report

The manuscript “Nanocomposites for X-ray photodynamic therapy” by Zaira Gadzhimagomedova et al. is an interesting and compact review of the different nanoparticles and/or nano-structured materials used or suitable for photodynamic therapeutic applications.

The text in the present form is not suitable for publication and a list of comments/suggestions is given below.

Page 2/14 - line 52. Authors introduce the Förster Energy Resonance Transfer (FRET). A reference is request.

Page 3/14 - line 72 "Cells were irradiated by x-ray for 150 cGy"

It is very important to add information also on the modality of irradiation, e.g., energy and time. The energy was monochromatic with a certain bandwidth? The irradiation was continuous or pulsed?
These details are important as the definition of the dose you are using in the text. Is this the Absorbed dose measured in Gy? Why in other part of the manuscript you are referring to the dose rate in term of Gy/min or kVp?
In a review paper it would be useful for the reader to better understand the type of "dose" and possibly use the same unit for comparison.
Alternatively the explanation of the different values of dose will be extremely useful to readers.

Page 3/14 - lines 102/104 "Nanoparticles n-Bu4N)2(Mo6I8(OCOCF3)6), efficiently absorbed X-rays because they consist of heavy elements, which in turn leads to the generation of the excited triplet states that interact with 103 molecular oxygen"
What are the triplet states involved in the process? Please clarify.

Page 3/14 - lines 107/109 "Thus, enhancement effectiveness of the n-Bu4N)2(Mo6I8(OCOCF3)6) nanoparticles could be achieved by their size."
What does it mean this sentence?

Page 5/14 - lines 182/183 "The use of this nanoparticles demonstrates increased antitumor therapeutic effectiveness."
How the therapeutic response is measured? It is possible to give some figures in these experiments and not just be simply qualitative?
The same question holds at page 6/14 - lines 191/192 "So, the X-PDT treatment minimizes the potential adverse effects of local radiotherapy"
Is it possible to clarify the advantages?

Page 6/14 - line 205 "...these nanoparticles convert X-ray energy due to their ability to stop ionizing radiation and...."
Are you sure NPs stop ionizing radiation? I think they "absorb" a significant fraction of the ionizing radiation. Isn't it?

Page 6/14 - lines 219/220 "Studied nanocomposites may be excellent candidates for the application in XPDT."
Why these nano composites are good candidates?

Page 7/14 - line 283 "...Hf-DBB-Ru was irradiated with 0, 1, 2, 3, 5 or 10 Gy X-ray."
I think "O Gy" corresponds to no irradiation. Is it correct?

Page 7/14 - line 287/289 "Mitochondria-targeted particles depolarize the mitochondrial membrane to activate apoptosis of cancer cells, which results in significant regression of colorectal tumor cells."
What is the range of doses to observe this effect?

Page 7/14 - lines 295 "Furthermore, Hf is a safe agent without considerate biotoxicity"
What does it mean this sentence?

Page 8/14 - lines 332/333 and 341/343 "All these challenges have rendered a great many ROS-based anti-cancer approaches ineffective [76]."
"For instance, type I PDT technology is capable of inducing ROS generation even in hypoxia [23]; [59] thus turning cancer stem cells sensitive to treatment along with transit amplifying cells."
What these two sentences mean? Please clarify.

Finally two additional requests:

1) Conclusions - They are really too general. Authors could add some figures related to applications such as energy, doses, reliability, efficacy, etc...

2) The technique here presented is certainly suitable for deep lying tumors, but are available evaluations/comparisons in case where both XPDT and PDT can be either applied?

Acronyms. Please check all acronyms and/or introduce/explain at its first appearance, e.g., FRET, ROS, ??

Finally a careful revision of both grammar and language is mandatory. The text submitted contains several mistakes that make the contribution hard to read and need to be fixed.
As an example, page 1
Line 43 " of a define wavelength."
line 44 "wirelenght" ?
page 3 - line 61 "Active core of nanoparticles sometimes was coated with silica cover"
etc...

Author Response

Response to Reviewer comments
ijms-767419

May 19, 2020

Dear Reviewer,

Thank you very much for your detailed comments! It is very helpful for us. Below, under your comments, we present our explanations and changes we have done according your advices.

Point 1 (page 2/16 - line 55-56): Authors introduce the Förster Energy Resonance Transfer (FRET). A reference is request.

Response 1: Reference was added: “But there are photoconverting nanoparticles based on the Förster resonance energy transfer [29]2”

Point 2 (page 3/16 - line 73) "Cells were irradiated by x-ray for 150 cGy"

It is very important to add information also on the modality of irradiation, e.g., energy and time. The energy was monochromatic with a certain bandwidth? The irradiation was continuous or pulsed?

Response 2: Unfortunately, more detailed information was not provided in the paper we are discussing. Not all cited articles provide fully detailed description (energy, time and so on). However, we tried to give in the revised manuscript more detailed information where is it possible.

Point 3: These details are important as the definition of the dose you are using in the text. Is this the Absorbed dose measured in Gy? Why in other part of the manuscript you are referring to the dose rate in term of Gy/min or kVp?

In a review paper it would be useful for the reader to better understand the type of "dose" and possibly use the same unit for comparison. Alternatively the explanation of the different values of dose will be extremely useful to readers.

Response 3: Thank you for this important comment. Now, only absorbed dose (in Gy) is used in our manuscript making it easy to read.

Point 4 (page 3/16 - lines 101/103): "Nanoparticles n-Bu4N)2(Mo6I8(OCOCF3)6), efficiently absorbed X-rays because they consist of heavy elements, which in turn leads to the generation of the excited triplet states that interact with 103 molecular oxygen" What are the triplet states involved in the process? Please clarify.

Response 4: Thank you! Excited triplet states of photosensitizers are involved in the process. We have done corresponding changes to the manuscript. (But authors of the paper did not provide more detailed information of the process.)

Point 5 (page 3/16 - lines 107/1011): "Thus, enhancement effectiveness of the n-Bu4N)2(Mo6I8(OCOCF3)6) nanoparticles could be achieved by their size." What does it mean this sentence? 

Response 5: The small size of the nanoparticles provides a large energy transfer. The smaller the particles are, the closer they can approach the target. This means that the value of the transferred energy from the nanoparticle to molecular oxygen is higher. As a result, it leads to the formation of bigger number of ROS. Thus, enhancement effectiveness of (n-Bu4N)2(Mo6I8(OCOCF3)6) nanoparticles could be achieved by their size. The X-ray dose also was 1 Gy and 2 Gy for 100 s.

We have done corresponding changes to the manuscript.

Point 6 (page 5/16 - lines 209/213): "The use of this nanoparticles demonstrates increased antitumor therapeutic effectiveness."

How the therapeutic response is measured? It is possible to give some figures in these experiments and not just be simply qualitative?

Response 6: The volume of tumor (HeLa cell line was used) with introduced LSZNPs and exposed to X-ray radiation of 8 Gy was almost completely inhibited in 15 days. In addition, late-stage apoptosis featuring karyopyknosis, karyorrhexis, and significant formation of apoptotic bodies was observed after irradiation of the tissues. Thus, the application of these nanoparticles demonstrate antitumor therapeutic effectiveness.

We have done corresponding changes to the manuscript.

Point 7 (page 5-6/16 – lines 220/235): "So, the X-PDT treatment minimizes the potential adverse effects of local radiotherapy" Is it possible to clarify the advantages?

Response 7: So, XPDT treatment minimizes the potential adverse effects of local radiotherapy due to the application of low radiation doses 1 Gy, while the typical dose for a solid epithelial tumor varies from 60 to 80 Gy.

Point 8 (page 6/16 - line 247/248): "...these nanoparticles convert X-ray energy due to their ability to stop ionizing radiation and...." Are you sure NPs stop ionizing radiation? I think they "absorb" a significant fraction of the ionizing radiation. Isn't it?

Response 8: We agree with your comment and corrected the sentence.

Point 9 (page 6/16 - lines 264-267): "Studied nanocomposites may be excellent candidates for the application in XPDT." Why these nano composites are good candidates?

Response 9: Emission bands of Tb3+ overlap well with the absorption lines of protoporphyrin PpIX. Therefore, the efficient energy transfer from donor to acceptor and the production of singlet oxygen were expected. This combination of nanoparticles generated ROS. Nanocomposites under investigation may be quite good candidates for the application in XPDTWe have done corresponding changes to the manuscript.

Point 10 (page 8/16 - line 353): "...Hf-DBB-Ru was irradiated with 0, 1, 2, 3, 5 or 10 Gy X-ray." I think "O Gy" corresponds to no irradiation. Is it correct?

Response 10: We agree with your comment and corrected this sentence.

Point 11 (page 8/16 - line 357/360): "Mitochondria-targeted particles depolarize the mitochondrial membrane to activate apoptosis of cancer cells, which results in significant regression of colorectal tumor cells." What is the range of doses to observe this effect?

Response 11: The 100−150 mm3 tumor was irradiated with energy of 180 J/cm2. We have done corresponding changes to the manuscript.

Point 12 (page 8/16 - lines 365/368): "Furthermore, Hf is a safe agent without considerate biotoxicity" What does it mean this sentence?

Response 12: In addition, series of experiments were carried out to check the cytotoxicity of HfO2 nanoparticles in [66]. It is reported that 3T3 fibroblast cell lines were used. Damage of the cell lines was not observed. Thus, the Hf is a biocompatible chemical element.

We have done corresponding changes to the manuscript.

Point 13 (page 9/16 - lines 420/433): "All these challenges have rendered a great many ROS-based anti-cancer approaches ineffective [76]." What these two sentences mean? Please clarify.

Response 13: ROS-based approaches rely on cancer cells killing due to oxidative damage of cellular components induced by ROS donors and redox-cyclers (chemical mode of action) or ROS inducers (signaling mode of action). Although effective against transit amplifying cancer cells, these do not affect cancer stem cells - which, along the epithelial-mesenchymal transition (EMT) process, are the primary target in cancer therapy. 

Cancer stem cells possess inherently increased antioxidant capacity, and most therapeutic agents (including those inducing ROS) are incapable of prolonged residing in cancer stem cells due to drug efflux pumps highly represented on the membrane of cancer stem cell. Additionally, just as normal stem cells, cancer stem cells localize to hypoxic niches, and deficient oxygen renders ROS generation intensity a priori low. Additionally, ROS inducers disregarding their mode of action promote strong antioxidant and anti-xenobiotic signaling further enhancing the cancer stem cells resistance to therapy.

Point 14 (page 9/16 - 441/488): "For instance, type I PDT technology is capable of inducing ROS generation even in hypoxia [23]; [59] thus turning cancer stem cells sensitive to treatment along with transit amplifying cells."

What these two sentences mean? Please clarify.

Response 14: Type I PDT agents are unique in this regard. The dependence of type I PDT on molecular oxygen levels is lower than that of the type II PDT if hydroxyl radical is generated, and, consequently, type I PDT allows oxygen-independent ROS generation. Since type I PDT agents do not require oxygen and do not need to enter or remain within a cancer stem cell to induce cell death (and thus, do not induce adaptive signaling changes in the cell), these agents do not have limitation characteristic of the most ROS-based agents for cancer therapy imposed by the above discussed facts. 

Point 15 (‘Finally two additional requests’):

1) Conclusions - They are really too general. Authors could add some figures related to applications such as energy, doses, reliability, efficacy, etc...

Response 15: We rewrote the conclusion, added specific radiation dose data, and compared the nanoparticles.

We have done corresponding changes to the manuscript.

Point 16:

2) The technique here presented is certainly suitable for deep lying tumors, but are available evaluations/comparisons in case where both XPDT and PDT can be either applied?

Response 16: Both XPDT and PDT can be applied to surface types of cancer, because optical beam could not penetrate deeper the several millimeters in the biological tissues. One should mention that x-ray radiation itself (even without photodynamic effect) could successfully treat some skin cancers [1, 2]. In spite of the XPDT is aimed mostly to treat deep tumor tissues where standard PDT could not be applied because of low penetration depth of optical photons, there are some trials to use XPDT for skin cancers as well. For example, cutaneous squamous cell carcinoma was found to be successfully treated by XPDT [3]. On the other hand, x-rays also damaged the surrounding normal cells, such as the keratinocytes and fibroblasts, which caused ulceration and hindered the healing of ulcerations [4]. Thus, it seems reasonable to split the regions for standard PDT and XPDT application (superficial and deep-lying tumor tissues, correspondingly).

  1. The Role of Kilovoltage X-rays in the Treatment of Skin Cancers, V Wolstenholme and J P Glees, EUROPEAN ONCOLOGICAL DISEASE 2006, 32- 35.
  2. Gianfaldoni S, Gianfaldoni R, Wollina U, Lotti J, Tchernev G, Lotti T. An Overview on Radiotherapy: From Its History to Its Current Applications in Dermatology. Open Access Maced J Med Sci. 2017 Jul 25; 5(4):521- 525.
  3. The effectiveness and safety of X-PDT for cutaneous squamous cell carcinoma and melanoma, Lei Shi , Pei Liu , Jing Wu , Lun Ma , Han Zheng , Michael P Antosh , Haiyan Zhang , Bo Wang , Wei Chen & Xiuli Wang, Nanomedicine (Lond.) (2019) 14(15), 2027–2043
  4. Iyer S, Balasubramanian D. Management of radiation wounds. Indian J. Plast. Surg. 45(2), 325–331 (2012).

We have done corresponding changes to the manuscript.

Thank you one more time for your thorough review of our manuscript.

Sincerely yours,

Gadzhimagomedova Zaira*
The Smart Materials Research
Institute of Southern Federal University,
Rostov-on-Don, Russia
e-mail: zaira31may@gmail.com

Zolotukhin Peter
Academy of Biology and Biotechnologies,
Southern Federal University, Rostov-on-Don, Russia
e-mail: pvzolotuhin@sfedu.ru

Kit Oleg
National Medical Research
 Centre of Oncology, Rostov-on-Don, Russia
e-mail: onko-sekretar@mail.ru

Kirsanova Daria
The Smart Materials Research
Institute of Southern Federal University,
Rostov-on-Don, Russia
e-mail: dkirs27@gmail.com

Soldatov Alexander
The Smart Materials Research
Institute of Southern Federal University,
Rostov-on-Don, Russia
e-mail: soldatov@sfedu.ru

* - Corresponding author

Reviewer 2 Report

The review is adequate for the subject matter. However, the style can be significantly improved to facilitate reading. For example, a table should be created to list all the materials, their QE, toxicity, ROS to be used, pros and cons for medical applications, or other comments. The table should be more informative than other tables available in the literature. 

A few papers should be cited, including a recent book on X-ray nanochemistry that discusses scintillators and those for medical applications. 

English can be improved too. "Can't" is not usually used in written English. Also the tone is too casual in many places. Be consistent when spell "X-ray(s)." Do not mix upper and lower cases. 

Author Response

Response to Reviewer comments
ijms-767419

May 19, 2020

Dear Reviewer,

Thank you for your comments related to the article! It has been helpful to all of us. Below, under your comment, we tried to explain and correct some aspects.

Point 1: The review is adequate for the subject matter. However, the style can be significantly improved to facilitate reading. For example, a table should be created to list all the materials, their QE, toxicity, ROS to be used, pros and cons for medical applications, or other comments. The table should be more informative than other tables available in the literature.

Response 1: This is an excellent idea! We focus on radiation doses in the article. Therefore, our table is as follows:

Table 1. Agents using in XPDT: PS without ScNP, PS with ScNP and MOF structure.

Photosensitizer

Scintillating nanoparticle

Absorbed dose of ionizing radiation (Gy)

Biological experiments

Reference

Cu-Cy

-

5 Gy for 30 min

Squamous cells carcinoma

[34]

TiO2:Ce

-

0.133 Gy for 100 seconds

A549

[35]

TiO2:C

-

0.133 Gy for 100 seconds

A549

[36]

(n-Bu4N)2(Mo6I8(OCOCF3)6)

-

1 and 2 Gy

HeLa and MRC

[37]

RB

CeF3:Tb3+, Gd3+

3 Gy

Mgc803, HEK293T and 4T1

[50]

ZnGa2O4:Cr

ZnPcS4

~0.18 Gy for 2 min

HeLa

[52]

ZnS:Cu, Co

TBrRh123

(tetrabromorhodamine-123)

2 Gy

PC3

[53]

 nMOF Hf -TCPP

Hf as radio-sensitizer;

6 Gy for 3 min

4T1, HeLa, and NIH3T3

[56]

nMOF

Hf6-DBA (Hf63-O)43-OH)4(DBA)6) and

Hf12-DBA (Hf123-O)83-OH)82-OH)6(DBA)9)

Hf as radio-sensitizer

2 Gy

CT26

[57]

Point 2: A few papers should be cited, including a recent book on X-ray nanochemistry that discusses scintillators and those for medical applications.

Response 2 (Page 1/16 line 37): “The principle of standard PDT is widely described in the existing literature [..]. In particular, it has been reported on the basic principle of PDT [..], Cerenkov Radiation mechanism [..], detailed consideration of scintillating nanoparticles and their role in the activation processes of photosensitizers [..] [1] as well as their medical applications – [2,3]”

  1. Guo, T. X-ray Nanochemistry: Concepts and Development; Springer: 2018; https://doi.org/10.1007/978-3-319-78004-7.
  2. Figueiredo, P.; Bauleth-Ramos, T.; Hirvonen, J.; Sarmento, B.; Santos, H.A. The Emerging Role of Multifunctional Theranostic Materials in Cancer Nanomedicine. In Handbook of Nanomaterials for Cancer Theranostics, 2018; 10.1016/b978-0-12-813339-2.00001-3pp. 1-31.
  3. Conde, J. Front Matter. In Handbook of Nanomaterials for Cancer Theranostics, 2018; 10.1016/b978-0-12-813339-2.09999-0pp. i-ii.

Point 3: English can be improved too. "Can't" is not usually used in written English. Also the tone is too casual in many places. Be consistent when spell "X-ray(s)." Do not mix upper and lower cases.

Response 3: We tried to take into account all comments and the manuscript has been edited.

We have done corresponding changes to the manuscript.

Thank you one more time for your thorough review of our manuscript.

Sincerely yours,

Gadzhimagomedova Zaira*
Affiliation: The Smart Materials Research
Institute of Southern Federal University,
Rostov-on-Don, Russia
e-mail: zaira31may@gmail.com

Zolotukhin Peter
Affiliation: Academy of Biology and Biotechnologies,
Southern Federal University, Rostov-on-Don, Russia
e-mail: pvzolotuhin@sfedu.ru

Kit Oleg
Affiliation: National Medical Research
 Centre of Oncology, Rostov-on-Don, Russia
e-mail: onko-sekretar@mail.ru

Kirsanova Daria
Affiliation: The Smart Materials Research
Institute of Southern Federal University,
Rostov-on-Don, Russia
e-mail: dkirs27@gmail.com

Soldatov Alexander
Affiliation: The Smart Materials Research
Institute of Southern Federal University,
Rostov-on-Don, Russia
e-mail: soldatov@sfedu.ru

* - Corresponding author

Round 2

Reviewer 1 Report

The manuscript has been improved. 

Many sentences have been now fixed and the text is certainly adequate to a general reader. However, for a review manuscript a bit of improvements are necessary:

a) language. A revision of many sentences has to be performed. 

Just to mention one sentence at page 4: ".... it leads to the formation of bigger number of ROS....."  I suggest to replace with ".... it leads to the formation of a higher number of ROS....."  

Please check.

b) I strongly recommend to revise the conclusion. As underlined in the first revision:

"Conclusions are really too general. Authors could add some figures related to applications such as energy, doses, reliability, efficacy, etc..."

In addition, in the abstract you written: "The role of metal-organic frameworks for applying XPDT as well as the mechanism underlying the generation of reactive oxygen species are discussed." where the mechanism is discussed in the revised text?

Author Response

Response to the Reviewer’s comments (Round 2)
ijms-767419

May 27, 2020

Dear Reviewer,

Thank you very much for your comments! Below, under your comments, we present explanations and changes we have done accordingly to your advice.

Point 1: a) language. A revision of many sentences has to be performed.

Just to mention one sentence at page 4: ".... it leads to the formation of bigger number of ROS....."  I suggest to replace with ".... it leads to the formation of a higher number of ROS.....".

Response 1: Thank you! The manuscript will be edited.

Point 2: b) I strongly recommend to revise the conclusion. As underlined in the first revision:

"Conclusions are really too general. Authors could add some figures related to applications such as energy, doses, reliability, efficacy, etc..."

Response 2: We rewrote conclusions according to your comment:

“Therefore, in this article various types of nanoparticles, photosensitizers, and their combinations were considered. For instance, there is an interesting combination of ZnGa2O4:Cr with ZnPcS4 which is activated by low doses of X-ray radiation (~0.18 Gy for 2 min). Moreover, some nanoparticles combine properties of both photosensitizers and scintillating nanoparticles. These nanoparticles are activated by rather low doses of irradiation in the range from 0,133 to 8 Gy (see details in Table 1). For example, TiO2:Ce and TiO2:C were irradiated with an X-ray dose of 0.133 Gy for 100 seconds. These doses are much less than in other cases. For instance, (n-Bu4N)2(Mo6I8(OCOCF3)6) and ZnS:Cu, Co/TBrRh123 were exposed to 2 Gy. With regard to MOFs, Hf-DBB-Ru nMOFs in the tissue were irradiated by 2 Gy (in energy 180 J/cm2 or 650 nm). The heavy metal makes MOFs promising candidates for X-ray photodynamic therapy. However, radiation doses for activation of MOF structures are higher than the dose for TiO2:Ce and TiO2:C.”

Point 3: In addition, in the abstract you written: "The role of metal-organic frameworks for applying XPDT as well as the mechanism underlying the generation of reactive oxygen species are discussed." where the mechanism is discussed in the revised text?

Response 3: Thank you for this important comment. We wrote some information about the ROS generation mechanism.

          “Reactive oxygen species (ROS) are chemically reactive chemical species containing oxygen. Examples include peroxides, superoxide, hydroxyl radical, singlet oxygen, and alpha-oxygen. There are several sources and corresponding mechanisms of ROS generation. One could classify them into two main groups: endogenous and exogenous sources. ROS are produced during a variety of biochemical reactions outside and within the cell - in the cytosol and all organelles (most notably in mitochondria, peroxisomes, and endoplasmic reticulum). Endogenous sources and mechanisms can be stimulated by external agents - these influences can also be classified as exogenous ROS inducers.  Formation of ROS can be stimulated by a variety of agents such as pollutants, heavy metals, tobacco, smoke, drugs, xenobiotics, or radiation. One could find an extensive review of the ROS sources and corresponding mechanisms in a recent review by Snezhkina et al [1].”

  1. Snezhkina, A.V.; Kudryavtseva, A.V.; Kardymon, O.L.; Savvateeva, M.V.; Melnikova, N.V.; Krasnov, G.S.; Dmitriev, A.A. ROS Generation and Antioxidant Defense Systems in Normal and Malignant Cells. Oxidative Medicine and Cellular Longevity 2019, 2019, 17, doi:https://doi.org/10.1155/2019/6175804.

We have done corresponding changes to the manuscript.

Thank you one more time for your thorough review of our manuscript.

Sincerely yours,

Gadzhimagomedova Zaira*
The Smart Materials Research
Institute of Southern Federal University,
Rostov-on-Don, Russia
e-mail: zaira31may@gmail.com

Zolotukhin Peter
Academy of Biology and Biotechnologies,
Southern Federal University, Rostov-on-Don, Russia
e-mail: pvzolotuhin@sfedu.ru

Kit Oleg
National Medical Research
 Centre of Oncology, Rostov-on-Don, Russia
e-mail: onko-sekretar@mail.ru

Kirsanova Daria
The Smart Materials Research
Institute of Southern Federal University,
Rostov-on-Don, Russia
e-mail: dkirs27@gmail.com

Soldatov Alexander
The Smart Materials Research
Institute of Southern Federal University,
Rostov-on-Don, Russia
e-mail: soldatov@sfedu.ru

* - Corresponding author